# Some of the Physical and Mechanical Properties of Particleboard Made from Betung Bamboo (*Dendrocalamus asper*)

**Lina Karlinasari** [1,*]**, Prabu Setia Sejati** [2]**, Ulfa Adzkia** [1]**, Arinana Arinana** [1] **and Salim Hiziroglu** [3]

[1] Department of Forest Products, Faculty of Forestry and Environment, IPB University,
Jl. Lingkar Akademik, Darmaga, Bogor 16680, West Java, Indonesia; ulfa23adzkia@apps.ipb.ac.id (U.A.);
arinana@apps.ipb.ac.id (A.A.)

[2] Research Center for Biomaterials, Indonesian Institute of Sciences, Jl. Raya Bogor Km. 46,
Bogor 16911, West Java, Indonesia; prab002@lipi.go.id

[3] Department of Natural Resource Ecology and Management, Oklahoma State University,
Stillwater, OK 74075, USA; salim.hiziroglu@okstate.edu

**\*** Correspondence: karlinasari@apps.ipb.ac.id

**Abstract:** The objective of this study was to evaluate various physical and mechanical properties of experimental particleboard panels made from Asian giant bamboo (*Dendrocalamus asper*). Single layer panels having a density level of 0.75 g/cm³ from coarse and fine particles were used within the scope of this study. Thickness swelling, water absorption, surface roughness, and wettability characteristics of the samples were tested as physical properties while bending, internal bond strength, and screw withdrawal strength of the panels were considered for their mechanical properties. Resistance of the panels against termite and fungus were also determined. Based on the findings in the work both physical and mechanical properties of the panels made from coarse particles resulted in higher values than those made from fine particles with the exception of their internal bond strength. It appears that using fine particles in the panels enhanced their overall surface quality as well as wettability. Regarding biological deterioration of the samples, those made with coarse particles had better resistance. It seems that giant bamboo as a non-wood lignocellulosic species would have potential to be used as raw material to the manufacture value added particleboard with accepted characteristics.

**Keywords:** asian giant bamboo; coarse particle; fine particles; surface roughness; biological deterioration

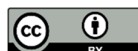

## 1. Introduction

It has been reported that there are currently more than 121 bamboo genera encompassing about 1662 species in the world [1]. Bamboo has a relatively long history of use as a building component, whether in the form of the culm or as a raw material of composite products. As of January 2021, there were 116 and 222 publications with the keywords "particleboard" and "bamboo", respectively, from the Scopus and Science Direct databases. The number of publications has more than doubled, compared with a report by [2], which found 53 publications as of 2016. Furthermore, a search using the keyword "particleboard" paired with the bamboo species *Dendrocalamus asper* yields 11 publications in the Scopus database and 10 in Science Direct. These publications generally discuss the basic physical and mechanical properties of bamboo composite panels as the function of treatments and type of adhesive used.

*Dendrocalamus asper (Schult.f.) Backer* is a bamboo species with a wide distribution in tropical and subtropical areas, especially in Southeast and South Asia [3]. This species is

referred to by various names, including betung, petung, batuang, and betho [3] which has also been introduced in many areas of the world including Latin and South America [2]. Commonly known as giant bamboo or rough bamboo, the bamboo is tall having large culms when it is mature at the age of 3 to 5 years. The culms can reach as high as 20 to 30 m with a diameter of 8 to 20 cm with 40 cm of internode spacing [4]. The wall thickness of the culm is around 6 to 22 mm, with a density of 0.52 to 0.56 g/cm³ [4–7]. Due to its thick and strong shells, *D. asper* culm is mainly used as a building component, in addition to the widely used manufacture for furniture, musical instruments, and household appliances [3]. The trend of using bamboo continues to increase and has even become a lifestyle, especially in Asia and some parts of South America [2]. In addition to the traditional uses of bamboo culm, bamboo has drawn interest for use in the manufacture of various composite panels including particleboard.

The particleboard made of bamboo has advantages in terms of the availability of the raw material, which comes from a natural resource being renewable, fast-growing, and environmentally friendly. Furthermore, bamboo particles can be used as a raw material, thus allowing the use of waste from bamboo production processes as well as bamboo waste from fields [8–11]. China, Malaysia, Costa Rica, India, and Vietnam have already developed bamboo particleboards at commercial scale [2,12].

Factors that affect the quality of bamboo particleboard are species, the age of the plants, the particle size, and the adhesive type [2]. Using bamboo as a raw material for panel manufacture has been discussed in previous studies [13–15]. Physical and mechanical properties of *D. asper* bamboo particleboards using small-size particles with various adhesive types have been investigated [7,11,16,17]. However, there is still a lack of information on different properties of particleboard panels manufactured from bamboo particularly their biological resistance including termites and decay as well as the surface quality. Previous studies carried out on *D. asper* culm assessed the resistance to subterranean termites [18,19], while Suprapti [20] investigated the resistance to fungal attack. The majority of particleboard is used as a substrate for thin overlays. Therefore, their surface quality in terms of surface roughness is an important factor for the overall quality of the final product [21]. Since there is limited information on the above areas of bamboo particleboard panels, the objective of this work was to evaluate the physical, mechanical properties, and biological resistance of experimental particleboard manufactured from bamboo so that such an underutilized non-wood resource can be considered as a raw material for the value-added product with a better understanding of their characteristics.

## 2. Material and Methods

### 2.1. Manufacture of Particleboard Panels

Approximately 3-year-old betung bamboo culms were collected in Lido, Bogor, West Java, Indonesia. The culms were cut manually into sections before they were converted into chips using a chipper. The chips were then dried and reduced into particles of two sizes, namely coarse and fine particles. For coarse particles, the dried chips were reduced into particles (10.10 ± 1.101) mm in length, (2.14 ± 0.344) mm in width, and (0.50 ± 0.015) mm in thickness employing a disc-flaker. Fine particles were manufactured using a hammer mill and screened on a 10-mesh sieve. All particles were then dried in an oven at 70 °C for 2 days to reach an air-dry moisture content of 3%–5%.

One cm thick, single-layer homogeneous panels were manufactured with a target density of 0.80 g/cm³. Isocyanate resin of diphenylmethane diisocyanate (MDI; Polyoshika Company, Jakarta, Indonesia) with 2% based on the weight of the oven-dried particles was used as a binder [22]. The particles and adhesives were mixed in a rotary drum mixer in a sealed environment. The mixture of particles and adhesive was then spread manually in a 300 × 300 mm (length × width) wooden frame. Metal bars were set against two parallel edges of the box as guides for reaching a board thickness of 1 cm after pressing. Compressing of the mats were carried out in laboratory press at a temperature of 150 °C using

a pressure of 2.5 N/mm² for 10 min. Pressed panels were conditioned in room temperature (25 °C) and 65% relative humidity for 2 weeks. Three replicate board samples were prepared for each particle size.

### 2.2. Physical and Mechanical Properties of the Panels

Physical properties of the panels were carried out based on the Japan Industrial Standard [23], including moisture content, density, thickness swelling (TS), and water absorption (WA) after 2 and 24 h of immersion in water. Static bending one-point loading of modulus of elasticity (MOE) and modulus of rupture (MOR), screw withdrawal resistance (SWR) on a wide surface, and internal bond (IB) of the samples were also evaluated based on the above standard for mechanical properties. The overall properties of the samples were also compared with the minimum requirements based on several standards, as displayed in Table 1.

**Table 1.** Minimum requirements for particleboards based on national and international standards.

| Physical and Mechanical Properties | Unit | Requirement | |
| --- | --- | --- | --- |
| | | **JIS A 5908 [23]** | **EN** |
| Thickness | mm | 5–13 | 13–20 [a] [24] |
| Moisture content | % | | 5–13 [b] [25] |
| Density | g/cm³ | 0.4–0.9 | - |
| Thickness swelling (24 h) | % | Max 12 | Max 14 [c] [26] |
| Bending strength = MOR | MPa | Min 8.0; 13.0 (Type 8; Type 13) | Min 11.0 [a(1)]; 13.0 [a(2)] [24] |
| MOE | GPa | Min 2,0; 2,5 (Type 8; Type 13) | Min 1600 [a(1,2)] [24] |
| IB | MPa | Min 0.15; 0.2 (Type 8; Type 13) | Min 0.24 [a(1)]; 0.35 [a(2)] [24] |
| SWP | N | Min 300; 400 (Type 8; Type 13) | - |

Notes: [a] Based on EN 312 [24], [a(1)] Type P1 (2005) for general uses, [a(2)] Type P2 (2005) for interior uses (including furniture), [b] EN 322 [25], [c] EN317 [26], MOR: Modulus of rupture; MOE: Modulus of elasticity; IB: Internal bond; SWR: Screw withdrawal resistance.

### 2.3. Surface Quality Measurement of the Panel

The surface quality of the panels was conducted by analysis of their surface roughness and wetting behavior. The samples of surface roughness were not sanded and directly measured using a portable stylus-type profilometer (Mitutoyo Surftest® SJ-210, Mitutoyo Corporation, Takatsu-tu City, Kanagawa, Japan). The detector had a diamond stylus with a tip radius of 5 μm. The average roughness ($R_a$), mean peak-to-valley height ($R_z$), and maximum peak-to-valley height ($R_y$) were used to evaluate the surface characteristics of the samples. Three different measurements were performed for each sample with a tracing length of 15 mm.

The wettability test of the samples was carried out to measure the contact angle (CA) between the water and the surface following the sessile drop [27]. About 0.03 mL of water was dropped onto the surface of the sample. The process of droplet spread and absorption from initial wetting until the water was absorbed into the sample's surface was video recorded using a high-resolution digital camera. The video was then cut every 10 s using the GOM Player software (Gretech Corporation, Seoul, Korea) for a total duration of 180 s.

The liquid drop on the samples surface was then measured using the digital image analysis software ImageJ (Wayne Rasband). The change in the CA as a function of time was calculated using a segmented regression model with a quadratic function in the beginning followed by a plateau function as a constant slope. The transition point between two functions was used to determine the equilibrium contact angle which plotted with a function of time. The equilibrium CA ($\theta e$) between the time (x) and the CA (y) was measured using the PROC NLIN function of SAS 9.4 Portable [27]. The K parameter was determined based on the S/G model [28]. The value of K is a constant indicating the reduction speed of the CA. A greater K value indicates that the wetting process takes a shorter time [29]. The K value of the S/G model is determined using the XLSTAT 2019 program (Addinsoft, Paris, France).

### 2.4. Resistance Evaluation of the Panels Against Fungus and Termite Exposure

The biological resistance of particleboard panels was carried out using the Indonesian National Standard (SNI) 7207-2006 [30] method regarding the resistance of wood products against wood-destroying organisms. The wood-destroying organisms, namely subterranean termites *Coptotermes curvignathus* Holmgren (Isoptera: Rhinotermitidae) and the wood-decay fungus *Schizophyllum commune* were used to evaluate the biological resistance of the samples.

Particleboard samples were cut to triplicate specimens of 25 × 25 × 10 mm. The test specimens were then put in an oven at a temperature of 60 ± 2 °C for 48 h and weighed to obtain the wood weight ($W_1$) before any tests were carried out. Fine sand as the substrate material and test bottles were sterilized separately by inserting them into a laminar flow hood for 48 h.

Each specimen was then inserted into a test bottle so that the broadest edge touched the wall of the bottle and the specimen was tilted. Then, 200 g of sterilized sand was put into the bottle to cover the specimens and 50 mL of water was added to keep the moisture of sand around 7%. A total of 200 healthy and active worker subterranean termites were introduced into each test bottle before it was closed with aluminum foil and stored in a dark room at 28 ± 2 °C and humidity of around 65% for 4 weeks. The activity of the termites in the bottle was weekly observed. After the 4 weeks period of time, the specimens were removed from the bottles and heated to a temperature of 60 ± 2 °C for 48 h to obtain the final weight ($W_2$). This weight was used to calculate the specimens' mass loss during the test. The percentage mass loss of the individual wood specimens was calculated by the difference in weights according to the following equation: % Mass loss = $(W_1 − W_2)/W_1$ × 100, where $W_1$ is the weight of oven-dried wood specimen before the test (g) and $W_2$ is the weight of oven-dried wood specimen after the test (g).

The particleboard specimens with dimensions of 50 × 25 × 10 mm were used for the decay test. All of the specimens were dried in an oven at a temperature of 103 ± 2 °C for 24 h to have their initial weights and then sterilized. The prepared fungal culture medium was made of potato dextrose agar. The closed petri dishes filled with a fungal culture medium were sterilized in an autoclave for 30 min with a pressure of 15 psi. Once sterile, the plate was placed flat so that the medium was at the bottom of the dish. The test fungus *S. commune* was inoculated onto the medium and incubated for a week. Next, sterile particleboard specimens were placed in petri dishes filled with an uncontaminated testing fungal culture. After 12 weeks, the exposed specimens were cleaned of the mycelium and heated in an oven at 103 ± 2 °C for 24 h before obtaining their final weight. Their mass loss was then calculated based on the weights before and after the test.

The average values of the physical-mechanical properties of the samples were compared with the minimum required by the norm of the standard, as stipulated in Table 1. Data analysis was assessed through the analysis of variance (ANOVA) with a further comparison of means by the Student's t-test ($p < 0.05$).

## 3. Results and Discussion

### 3.1. Physical Properties of the Panels

The density and moisture content of the bamboo chip raw material used in this study were $0.55 \pm 0.07$ g/cm$^3$ and $7.84 \pm 0.18\%$, respectively. The characteristics of the *D. asper* particleboards are summarized in Table 2.

**Table 2.** Average values (± SD) of the physical properties of *D. asper* particleboards.

| Particleboard | MC (%) | Density (g/cm$^3$) | CR |
|---|---|---|---|
| Fine particles | 8.59 ± 0.23 | 0.58 ± 0.10 | 1.05 ± 0.19 |
| Coarse particles | 9.03 ± 0.13 | 0.58 ± 0.03 | 1.05 ± 0.06 |

Note: Fine particles: Particles retained on a 10-mesh sieve; coarse particles: Particle dimension width ± 2 mm, length ± 10 mm, and thickness ± 0.5 mm; MC: Moisture content; CR: Compaction ratio.

In this study, the density of the *D. asper* particleboards was 0.58 g/cm$^3$, which was considered to be medium density (range 0.5–0.8 g/cm$^3$) [31] and was in line with Japan Industrial Standard (JIS) and EN standards. The particleboard with a medium density and thickness is commonly used in commercial products for panels, especially for furniture.

The density of the raw material is one of the major parameters influencing the overall panel density and properties. The relationship between the particleboard density and the raw material density is known as the compaction ratio (CR). In general, the desired CR value is >1.3 for medium-density particleboards. The CR value is related to the presence of sufficient contact areas among particles during the pressing, which results in good bonding and mechanical properties [31]. A low density for the raw material is generally the basic requirement for achieving the ideal CR value. The CR value for the two types of particleboards with different particle sizes was 1.05. This value was lower than the expected CR. The densification process may have been undermined by the loss of adhesive and particles during the manual preparation in the forming box or during the prepressing or hot-pressing process. This phenomenon can occur due to the loss of some adhesives and particles during manual preparation in the forming box or during the prepressing or hot-pressing process. The decrease in the density of both types of *D. asper* particleboards reached 40%, while the research conducted by [7] using the same type of bamboo with an urea-formaldehyde adhesive showed a decrease of 10%. In a past study, a decrease of about 30% of the target density with the use of citric acid adhesive was found [32].

Thickness swelling and water absorption are the key parameters in describing the dimensional stability of wood composites [33]. The WA and TS for each board are depicted in Figure 1. The results showed that the TS values of the samples were lower for the panels made with coarse particles compared with those made with fine particles for 2 and 24 h immersion (2.44% vs. 4.20% and 3.93% vs. 6.93%, respectively). Comparable results were found for the WA values, with lower values for boards made with coarse particles (27.35% at 2 h and 43.88% at 24 h) versus fine particles (39.25% at 2 h and 64.85% at 24 h). The ANOVA ($\alpha = 5\%$) showed that significant differences existed between the two particleboards made from different sized particles, for both TS and WA at 2 and 24 h. However, these TS and WA values met the requirements listed in JIS A 5908 and EN 317 for TS after immersion in water, as shown in Table 1. The use of isocyanate adhesive was previously proven to help particleboards have better water resistance properties [34]. The TS and WA of particleboards in the current study were better than those reported by [32] and [7] used citric-acid resin and urea-formaldehyde resin, respectively.

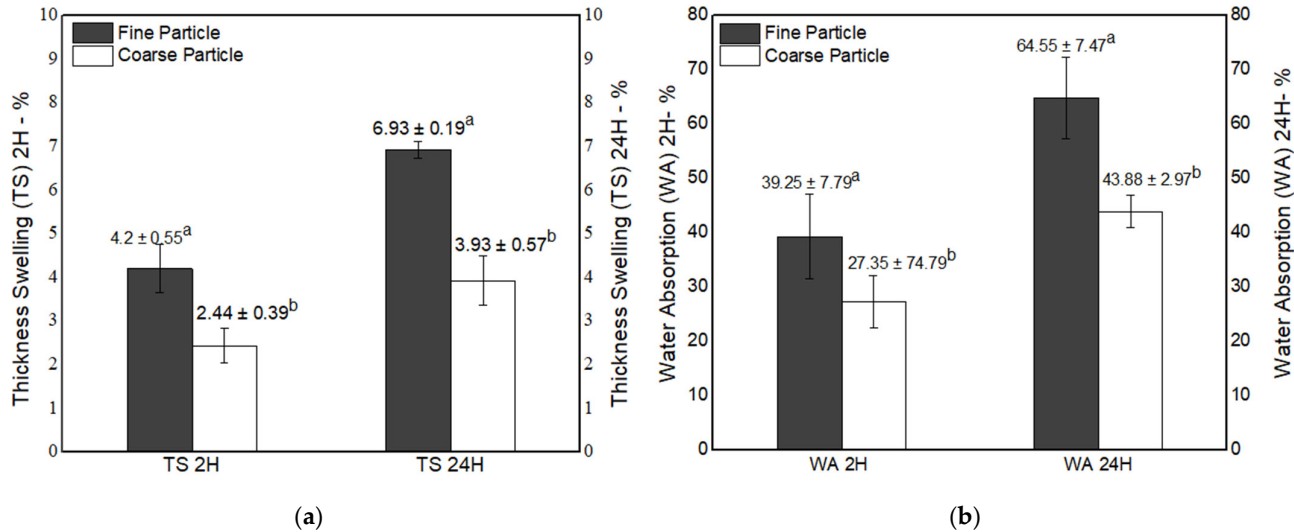

**Figure 1.** (**a**) Thickness swelling (TS); (**b**) water absorption (WA) of particleboards with different particle sizes. Values followed by different letters are statistically different at $p < 0.05$.

The particle size and adhesive type determine the particleboard's ability to attract water. The larger the particle size, the better the TS and WA physical properties. The difference in particle size and the use of MDI adhesive in this study led to decreases in TS by 35% and in WA by about 30%—both at 2 and 24 h—in boards made with coarse particles relative to those made with fine particles. A previous study carried out by Widyorini et al. [32] on single-layer particleboard using *D. asper* with a citric acid resin containing 15% adhesive, showed a decrease of about 8% for TS at 24 h from −10 mesh (fine particles) to −4 + 10 mesh (coarse particles). The TS and WA values (at 2 and 24 h) in this study were better than those of [7], who investigated the particleboard made from *D. asper* and urea-formaldehyde synthetic adhesive with a resin content of 10%. In that study, the particle size ranged from −35 + 40 mesh to −60 + 65 mesh and the particleboard was made based on −35 + 40 mesh particles alone and combined with particles of other sizes. In general, the composite boards made with coarser particles have smaller interparticle voids or passageways, and the WA is therefore slow throughout the 24 h test [31]. This phenomenon can also be found in particleboards made with fast-growing wood raw materials [35], which reveals that the coarser the particles, the lower the TS and WA values. The opposite condition is found for the particleboard made from agricultural materials, such as sunflower seed husks, where the coarser the particles, the higher the TS and WA values [36]. Such findings can be explained by the type of raw material and also have a role in determining the characteristics of particleboard behavior with regards to attracting water and affecting the uniformity of the board density [34, 36–41].

*3.2. Mechanical Properties of the Panels*

Bending properties of the panels, namely MOE and MOR are illustrated in Figure 2. Both MOE and MOR values of the panels made from coarse particles were higher than those made from fine particles. The MOE values for the boards made with coarse and fine particles were 1.49 and 1.31 GPa, respectively. Meanwhile, the MOR values, which indicated the bending strength, were 13.43 MPa for coarse particles and 11.37 MPa for fine particles. Panels made with coarse particles had IB values of 0.33 MPa, while the corresponding value was 0.57 MPa for those made with fine particles (Figure 3). Meanwhile, the SWR of coarse particleboard was 558 N, while that for fine particleboard was 441 N, as can be seen in Figure 4. The results of the ANOVA from the comparison test showed that the particleboard samples made with coarse and fine particles in this study were not significantly different for all the parameters of mechanical properties that were tested.

Referring to Table 1, the properties of the samples that met the JIS requirements [23] were the MOR and IB of coarse particleboard for its two standard types (Type 8 and Type 13), while fine particleboards met the MOR requirements for Type 8. Neither type of particleboard qualified for the MOE values of either Type 8 or Type 13 for over 2 and 2.5 GPa, respectively. Meanwhile, the particleboards in this study that met [25] were only coarse particleboards with regards to the MOR and IB properties. The MOR values for the coarse particleboard were close to the standard values, while all mechanical properties on the fine particleboard did not met the standard. Both types of particleboard met the qualification for SWR as stipulated in the JIS standard.

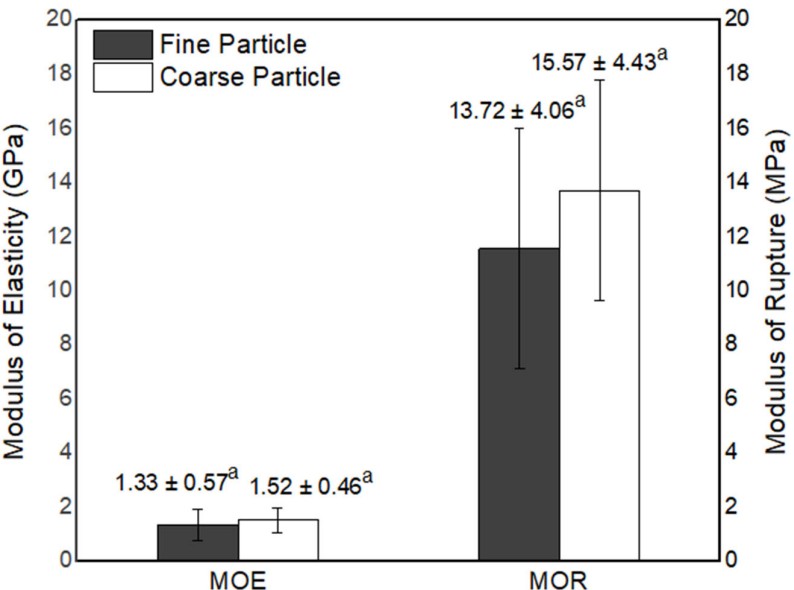

**Figure 2.** Bending characteristics of particleboard made with different particle sizes. Values followed by the same letters are statistically not different at $p < 0.05$.

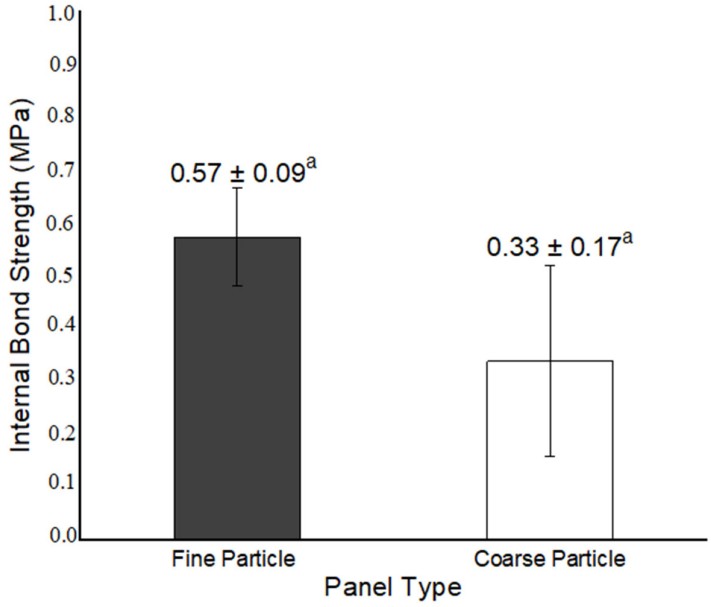

**Figure 3.** Internal bond strength values of the samples. Values followed by the same lowercase letters are statistically not different at $p < 0.05$.

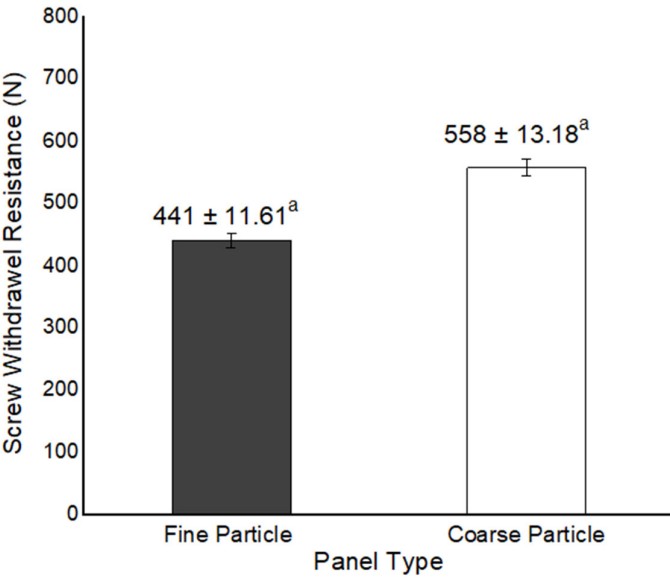

**Figure 4.** Screw withdrawal resistance of the sample. Values followed by the same letters are statistically not different at *p* < 0.05.

The particle size and geometry are two important raw material characteristics influencing almost all properties of the panels including IB and screw withdrawal resistance of the panel [42]. A study on the IB mechanical properties in bamboo particleboards found that the IB values were higher with a decreased particle size [7]. Another study on wood particleboard also determined similar results [43]. In this work, it was also observed that boards made with coarse particles had lower IB values than those with fine particles. Internal bond strength values of 0.57 and 0.33 MPa were determined for the panels made from fine and coarse particleboards, respectively. The study on composite boards made from veneer waste and having particle sizes of 0.5 to 2.0 mm had an IB value of 0.58 N/mm² [11], while [7] found an IB value of 0.25 N/mm² for the particleboard with a particle size of −35 + 40 mesh. In a comparison to the IB values from commercial bamboo particleboards, [2] found a range of IB values from 0.34 to 1.50 N/mm² for boards using various resin and resin contents. However, the SWR values on the face surface of the boards were higher for the coarse compared with the fine particleboard. Using the same bamboo species of *D. asper* with different particle sizes, [11] and [7] found the SWR (face) values of 499 and 706 N, respectively.

### 3.3. Surface Quality Analysis

Table 3 presents the surface roughness and CAs of the test samples. The fine particleboard had a smoother surface (6.20 μm ($R_a$), 7.79 μm ($R_q$), and 30.85 μm ($R_z$)) and higher CA at 0 s (104.4°) and k (0.011), compared with the coarse particleboard, which had a rougher surface (7.48 μm ($R_a$), 9.02 μm ($R_q$), and 32.48 μm ($R_z$)), lower CA (94.13°), and k (0.021). The representative example for measurement of the CA of a water droplet is shown in Figure 5. Dynamic contact angle measurements were plotted to show the evolution of contact angle overtime (Figure 6). The equilibrium CA (θe) of the regression model fit found that the contact angle exhibited an exponential decay becoming fairly stable for time (t) > 30s. The average θe of the fine particleboard is 88.52 s, while the coarse particleboard is 63.91 s. The reducing contact angle is related to a dynamic wetting process [28,44].

Certain parameters including particle compaction, fine screen particles, lower extractive content, higher press time, and moisture content play an important role in increasing the smoothness of the surface [45,46]. The current study found that the higher the surface

roughness or the coarser the particleboard, the lower the CA. In a previous study, wood species with a porous structure was found to have a higher surface roughness [47,48]. In terms of the coarser particleboard, the lower CA indicated perfect wetting, which means better hydrophilicity. It was also pointed by the k value, where the coarser particleboard had higher k values compared to the fine particleboard. The k corresponds to the easiness of the liquid spreads and penetrates [28,46]. This parameter is important in a gluing and coating system. A study by Shah [48] found that a surface of laminated bamboo revealed the CA was about 60° at 0 s and the starting table for time was more than 20 s in a wetting process. The time difference of reducing contact angle was due to the ability of water as a polar liquid to spread on the surface bamboo and be absorbed into the bamboo cellular structure [48].

**Table 3.** Average surface roughness and contact angles of *D. asper* particleboards.

| Particleboard | $R_a$ (μm) | $R_q$ (μm) | $R_z$ (μm) | CA (°) at 0 s | k |
|---|---|---|---|---|---|
| Fine particles | 6.20 [a] (±0.17) | 7.79 [a] (±0.28) | 30.85 [a] (±1.66) | 104.04 [a] (±15.25) | 0.011[a] |
| Coarse particles | 7.48 [b] (±0.47) | 9.02 [b] (±0.86) | 32.48 [a] (±4.18) | 94.13 [b] (±10.90) | 0.021[a] |

$R_a$: Arithmetical mean roughness; $R_q$: The root mean square value of the ordinate values; $R_z$: The maximum profile height; CA: Contact angle; k: Constant rate of change of the contact angle. Symbol ± in bracket denotes standard deviation values. Numbers followed by the same letters are not significantly different.

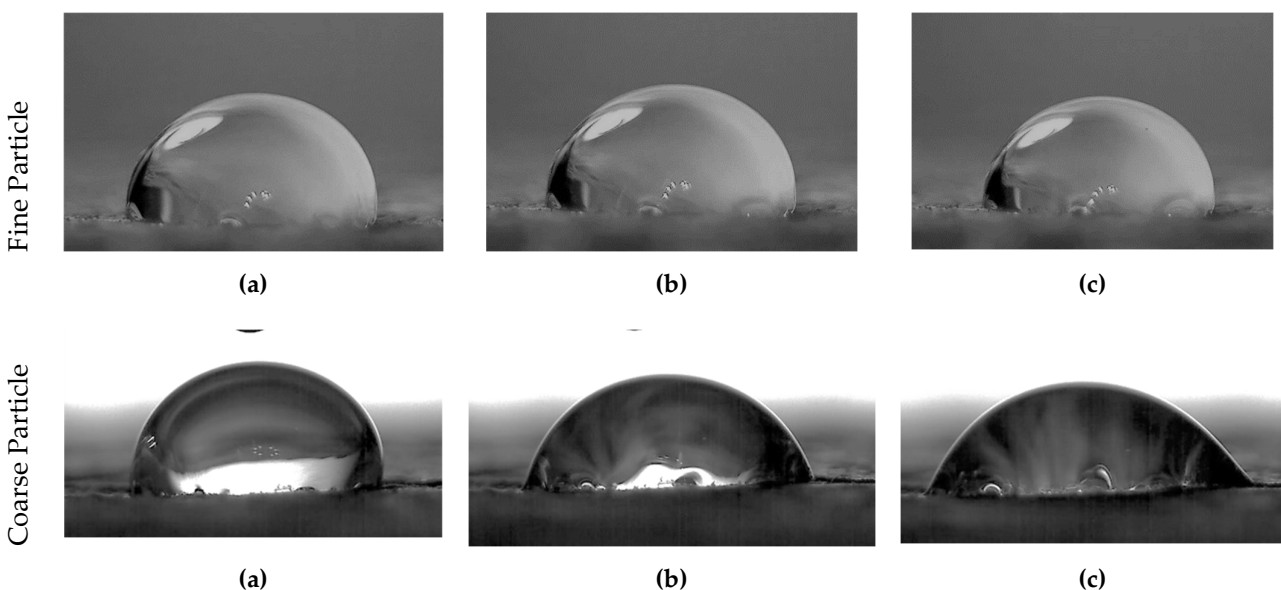

**Figure 5.** Representative measurement of the contact angle of a water droplet (**a**) the first time the water touches the surface of the board (t = 0 s); (**b**) as the water is absorbed (t = 60 s); (**c**) the static contact angle on fine and coarse *D. asper* bamboo particleboard (t = 180 s).

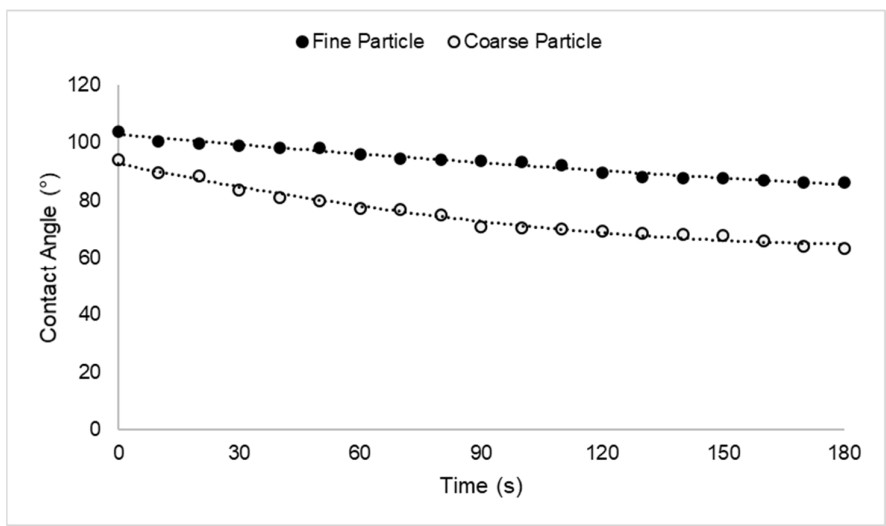

**Figure 6.** The dynamic wetting process as shown by the contact angle (°) with time.

### 3.4. Biological Deterioration of the Samples

Figure 7 shows that when the specimens were exposed to subterranean termites (*C. curvignathus)*, higher mass losses occurred in fine particleboards (12.97% ± 1.88) compared with coarse particleboards (12.41% ± 0.31). The attack by the white-rot fungus *S. commune* caused a higher mass loss from coarse particleboards (5.99% ± 0.64) than from fine particleboards. (5.48% ± 2.97). The higher standard deviation on mass loss subterranean termite denoted that the termite consumption rate was higher and varies than the fungi on lignocellulosic-based composite materials. The ANOVA revealed a significant difference between coarse and fine particleboards for termite resistance, while there were no significant differences for fungal attack. The different responses to termite and white-rot attack seemed to be related to the particle size.

The higher susceptibility of fine particleboard compared with coarse particleboard was presumably due to the coverage of the particles by the resin. Maloney [31] explained that the amount of adhesive used in the fine particleboard was not sufficient to cover the entire surface, so the contact between the particles was not optimal. This may have allowed a more severe attack by termites. In a previous study, the subterranean termite *C. curvignathus* attack led to a high mass loss (about 12%) for both fine and coarse bamboo (*D. asper*) particleboards, compared with a mass loss of about 5%–6% from particleboards made from fast-growing wood species, including jabon (*Anthocephalus cadamba*), sungkai (*Peronema canescens*), and mangium (*Acacia mangium*), and urea-formaldehyde adhesive resin [49]. *Coptotermes* spp. from Rhinotermitinae are recognized as one of the most important and dominant subterranean termite pests based on damages on building construction [50]. *Coptotermes curvignathus* is a termite that is associated with severe attack, mostly in Indonesia. It is not only a pest in timber structures, but also in oil palm plantations [51]. Therefore, the high susceptibility to subterranean termite attack is considered one of the main limitations to the use of bamboo as a raw material for various purposes.

In contrast, the fungal attack was more severe in coarse particleboards than fine ones. As for physical and mechanical properties, the relationship between particle size and the adhesive bonding system also affected the overall biological resistance properties of the samples. The adhesive in the panels would be responsible to influence the ability of the mycelium to degrade the wood component. The white-rot fungus *S. commune* can degrade chemical components, especially lignin in *D. asper* bamboo [52,53]. Moreover, white-rot fungi in general cause most severe attacks, with different effects for all the main chemical constituents, especially cellulose, polyoses, and lignin [54].

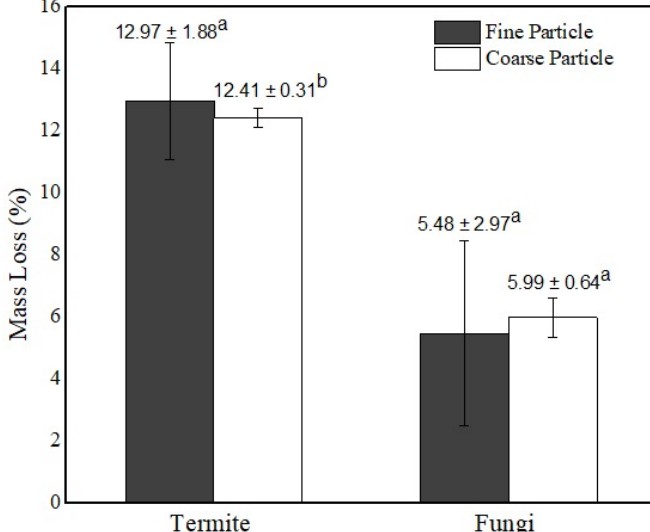

**Figure 7.** Mass loss of *D. asper* particleboards caused by subterranean termites (*C. curvignathus*) and white-rot fungus (*S. commune*). Values followed by different lowercase letters are statistically different at $p < 0.05$.

## 4. Conclusions

Based on the findings in this work, experimental particleboard panels made of coarse particles of bamboo resulted in better properties than that those made of fine particles in terms of their physical, mechanical, surface quality, as well as termite resistance properties with the exception of IB and fungal resistance characteristics. Excluding the MOE values, all properties of the specimens met the requirements listed in the Japanese and European standards for the particleboard for general interior use. Although having a relatively rough surface of the panels made with the coarse material had good wettability, which is necessary for adhesion and coating systems. It appears that even coarse particles of bamboo would have potential to be used as raw material to produce value-added panel products. In further studies, overlaying properties of such panels can be investigated to have a better understanding of their utilization as a substrate for different applications.

**Author Contributions:** Conceptualization, drafting, writing, L.K.; review, editing, supervision, L.K. and S.H.; Sample collection and preparation, laboratorium work, P.S.S.; Laboratorium work, original draft preparation, U.A.; Formal analysis, A.A. All authors have read and agreed to the published version of the manuscript.

**Funding:** This research was funded by the Indonesia Ministry of Research and Technology (RISTEK)/National Research and Innovation Agency (BRIN) through Research Grants for World Class Research entitled "High Performance Acoustical Engineered Composite Product from Indonesian Bamboo" FY 2021.

**Data Availability Statement:** Not applicable.

**Acknowledgments:** The authors are grateful for the support of the Indonesia Ministry of Research and Technology (RISTEK)/National Research and Innovation Agency (BRIN) through Research Grants for World Class Research entitled "High Performance Acoustical Engineered Composite Product from Indonesian Bamboo" FY 2021.

**Conflicts of Interest:** The authors declare no conflict of interest.

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
