# Peer review of "Some of the Physical and Mechanical Properties of Particleboard Made from Betung Bamboo (Dendrocalamus asper)"

_applsci, doi:10.3390/app11083682_

Round 1
Reviewer 1 Report
Materials and methods
There is
No information on size distribution of particles is given for both types. It is expected to provide the analysis of length, width and thickness for representative quantity of particles, not mentioning other shape-based parameters of chips. These are determinants form various properties of panels.
Multiple standards are available on 2d surface roughness determination. Please point the one you used.
- 126: “…CA as a function of time was calculated using a polynomial regression model until a quadratic equation was obtained” – how to understand it? Additionally neither the regression model nor the equation is given in the result section.
Results and Discussion
The panels’ density issue needs to be corrected/explained. First the target density of 0.8 g/cm3 is given in section 2 (l. 88). Then in the table 2 two 0.58 values are displayed and in the line 181 even different density is mentioned 0.75g/cm3. What is it finally?
The 40% of the density decrease means that the laboratory scale manufacturing method should be improved, since you are not producing what you planned
I suggest to use another method of indicating values statistically equal or different than using “a” and “b” letters since for example figure 1 contains two plots marked also with “a” and “b”. This is confusing.
None of values of MOR and MOE (l.244-245) equals values displayed in figure 2! You should also use only one of MOE units: GPa or N/mm2
L.290 maybe “is” word is missing in the sentence?
l.295-296 the statement seems unjustified since in experiment only two types of panels were used. It is hard to generalize based on two cases.
l.298 “ideal wetting” – use more appropriate word
Table 3. delete “Degree” word following “CA”
Two figures 6 exist in the manuscript.
Fig. 6 (l. 349) – why the standard deviation is so different between fine and coarse particleboards?
Conclusions
l.354 correct the sentencje.
Reviewer 2 Report
The authors described the manufacture of single-layer homogeneous particleboard panels with a target density of 0.80 g/cm3 using different particle sizes/dimensions. The manuscript is well-written; however, some clarifications are needed before consideration for publication.
Comments:
- Line 185-187 on page 5, the authors stated that "the density of the raw material is one of the major parameters influencing overall panel density and properties. The relationship between the particleboard density and the raw material density is known as the compaction ratio (CR)". The authors should clarify that: 1.1. Is overall density of the single-layer particleboard associated with the particle size of raw material or associated with the density of the raw material? 1.2. Is the compaction ratio related to the particle size of the raw material or related to the density of the raw material? 1.3. The authors should reinforce the statement (line 185) with references.
- Line 318 on page 10, the authors need to check the figure number.
Reviewer 3 Report
Dear Authors,
I see your paper valuable for potential readers due to practical tests and useful and applicable results. But I have a few questions and comments .
1) How many samples were used for testing?
Line 96-97 "Three replicate boards samples were prepared for each particle size"
Does it mean there were only 3 test samples?
2) It would be appropriate to supplement Chapter 2.1 with a picture (scheme) of the composition of individual tested samples. Their composition is not entirely clear from the description.
3) Please briefly describe in chapter 2.2 the procedure for testing screw whitadrawal resistance, MOE and MOR. EU readers mostly don´t have access to the Japan Industrial Standard.
With regards!
Round 2
Reviewer 1 Report
attached

Author Response
Response the authors (comments to comments)
Some of The Physical and Mechanical Properties of Particleboard Made from Betung Bamboo (Dendrocalamus Asper)
Lina Karlinasari 1,*, Prabu Setia Sejati 2, Ulfa Adzkia 1, Arinana Arinana 1, and Salim Hiziroglu 3,*
|
No |
Comments of reviewers |
Comments from authors |
Comments of reviewers 2 |
Comments from authors (2) |
|
REVIEWER 1 |
|
|
|
|
|
|
Materials and methods |
|
|
|
|
1. |
No information on size distribution of particles is given for both types. It is expected to provide the analysis of length, width and thickness for representative quantity of particles, not mentioning other shape-based parameters of chips. These are determinants form various properties of panels. |
Thank you. We have explained the geometry and dimension of particle in paragraph 1 Ch. 2.1. |
At the scientific level giving three dimensions of a particle seems to be insufficient to describe a whole population. How many particles have You measured? One? Are they average values? What were the standard deviations? |
We could explain that for calculating the dimension of the particle, we measure the coarse particle size as size distribution in a number of 10 samples with the values with an average 10.10 mm (SD= +/- 1.101) in length, 2.14 mm (SD= +/- 0.344) in width, and 0.50 mm (SD= +/- 0.015) in thickness as result from disk flaker. Unfortunately, we did not measure the size of the chips as raw material after converting them from the bamboo culm. We also did not measure the fine particles, because we only put the chips to the hammer mill for making 10 mesh particle size.
We have revised with add explanation in the manuscript for the size distribution of particles in paragraph 1 Ch. 2.1. (yellow highlight)
|
|
2. |
Multiple standards are available on 2d surface roughness determination. Please point the one you used.
|
|
There was no answer for the comment. |
Thank you. We miss to answer
Currently there is no accepted standard method to evaluate surface roughness of PB panels |
|
126: “…CA as a function of time was calculated using a polynomial regression model until a quadratic equation was obtained” – how to understand it? Additionally neither the regression model nor the equation is given in the result section. |
The models were the way to obtain CA values. We have mentioned the reference for calculating and developing the model. We have added the explanation for the model in last sentence at Ch. 3.2.
[27] Darmawan, W.; Nandika, D.; Noviyanti, E.; Alipraja, I.; Lumongga, D.; Gardener, D. Wettability and bonding quality of ex-terior coatings on jabon and sengon wood surfaces. J. Coatings Technol. Res 2018, 15, 95–104 doi:10.1007/s11998-017-9954-1.
|
Sustain. Why general polynomial regression was used? Why not simply quadratic regression for the first segment of data? In the referenced paper it was clear from the beginning that the segmented regression function was applied: quadratic at the beginning and linear for later stage. |
We have revised and added the sentence to explain better about the method in determining CA with a segmented regression model in paragraph 3 Ch 2.3 (yellow highlight) |
|
|
|
Results and Discussion |
|
|
|
|
3. |
The panels’ density issue needs to be corrected/explained. First the target density of 0.8 g/cm3 is given in section 2 (l. 88). Then in the table 2 two 0.58 values are displayed and in the line 181 even different density is mentioned 0.75g/cm3. What is it finally? |
Thank you for you criticize. The target density was 0.8 g/cm3; and the actual density was 0.58 g/cm3 |
ok |
Thanks |
|
4. |
The 40% of the density decrease means that the laboratory scale manufacturing method should be improved, since you are not producing what you planned |
It is possible to happen as like also in others study. Thanks
|
Ok |
Thanks |
|
5. |
I suggest to use another method of indicating values statistically equal or different than using “a” and “b” letters since for example figure 1 contains two plots marked also with “a” and “b”. This is confusing. |
Thank you. We make a revision with “higher” letter for significancy. It also for others figure for make a clear. |
Have You? I still see it not corrected! |
Ok, thanks. We have revised the Figures to make tem clear and not confusing. |
|
6. |
None of values of MOR and MOE (l.244-245) equals values displayed in figure 2! You should also use only one of MOE units: GPa or N/mm2 |
We put MOE in GPa unit; and MOR in MPa unit as shown in Figure 2 |
??? You have not answered any of two concerns! 1. still different values in figure and in the text 2. MOE values still not unified (for example l.255 in version2. |
Thank you
1. I have changed the value in the text in N/mm2 to be GPa (see in text) 2. Yes, thanks for scrutinize Please see the Table 1 and paragraph 1 Ch 3.2 (the third last sentence, yellow highlight) |
|
7. |
L.290 maybe “is” word is missing in the sentence? |
Ok, thank you for your correction |
ok |
Thanks |
|
8. |
l.295-296 the statement seems unjustified since in experiment only two types of panels were used. It is hard to generalize based on two cases. |
We think it is clear as comparison for two kinds types of particles |
ok |
Thanks |
|
9. |
l.298 “ideal wetting” – use more appropriate word |
We change with “perfect wetting”
|
ok
|
Thanks |
|
10. |
Table 3. delete “Degree” word following “CA” |
Thank you for your correction |
ok |
Thanks |
|
11. |
Two figures 6 exist in the manuscript. |
Thank you we have made correction |
ok |
Thanks |
|
12. |
Fig. 6 (l. 349) – why the standard deviation is so different between fine and coarse particleboards? |
The termite consumption rate was higher than fungi on lignocellulosic-based composite materials, therefore the diversity of feeding capacity, which is described as the standard deviation becomes larger.
We have added the explanation in the text revision
|
ok |
Thanks |
|
|
Conclusions |
|
|
|
|
13. |
l.354 correct the sentence. |
We think it was right conclusion |
Ok. Sorry, my mistake. |
OK. Thanks |

Reviewer 2 Report
Considering the overall changes and responses that the authors have been provided, I would suggest this manuscript for publication.
Author Response
Response the authors (comments to comments)
Some of The Physical and Mechanical Properties of Particleboard Made from Betung Bamboo (Dendrocalamus Asper)
Lina Karlinasari 1,*, Prabu Setia Sejati 2, Ulfa Adzkia 1, Arinana Arinana 1, and Salim Hiziroglu 3,*
|
No |
Comments of reviewers |
Comments from authors |
|
REVIEWER 2 |
|
|
|
1. |
Line 185-187 on page 5, the authors stated that "the density of the raw material is one of the major parameters influencing overall panel density and properties. The relationship between the particleboard density and the raw material density is known as the compaction ratio (CR)". The authors should clarify that: 1.1. Is overall density of the single-layer particleboard associated with the particle size of raw material or associated with the density of the raw material? 1.2. Is the compaction ratio related to the particle size of the raw material or related to the density of the raw material? 1.3. The authors should reinforce the statement (line 185) with references. |
1.1. In related panel density, the density of the raw materials is important factor; the particle size become major factor influencing the board properties not only panel density 1.2 The CR related to density of raw material 1.3 We have put the reference for number [31]: Maloney, T.M. Modern Particleboard and Dry Process Fiberboard Manufacturing, Miller Freeman Inc: San Fransisco, 1993.
|
|
2. |
Line 318 on page 10, the authors need to check the figure number. |
Ok, thank you for correction. It should be Figure 7. |
